# Scabies outbreak management in refugee/migrant camps in Europe 2014–2017: a retrospective qualitative interview study of healthcare staff experiences and perspectives

Naomi A Richardson ,[1] Jackie A Cassell ,[1] Michael G Head ,[2] Stefania Lanza,[1] Corinna Schaefer,[3] Stephen L Walker ,[4,5] Jo Middleton [1,6]

► This web-only file has been produced by the BMJ Publishing Group from an electronic file supplied by the authors and has not been edited for content.

For numbered affiliations see end of article.

**Correspondence to**
Jo Middleton;
J.Middleton@bsms.ac.uk

## ABSTRACT

**Objectives** Provide insights into the experiences and perspectives of healthcare staff who treated scabies or managed outbreaks in formal and informal refugee/migrant camps in Europe 2014–2017.

**Design** Retrospective qualitative study using semistructured telephone interviews and framework analysis. Recruitment was done primarily through online networks of healthcare staff involved in medical care in refugee/migrant settings.

**Setting** Formal and informal refugee/migrant camps in Europe 2014–2017.

**Participants** Twelve participants (four doctors, four nurses, three allied health workers, one medical student) who had worked in camps (six in informal camps, nine in formal ones) across 15 locations within seven European countries (Greece, Serbia, Macedonia, Turkey, France, the Netherlands, Belgium).

**Results** Participants reported that in camps they had worked, scabies diagnosis was primarily clinical (without dermatoscopy), and treatment and outbreak management varied highly. Seven stated scabicides were provided, while five reported that only symptomatic management was offered. They described camps as difficult places to work, with poor living standards for residents. Key perceived barriers to scabies control were (1) lack of water, sanitation and hygiene, specifically: absent/limited showers (difficult to wash off topical scabicides), and inability to wash clothes and bedding (may have increased transmission/reinfestation); (2) social factors: language, stigma, treatment non-compliance and mobility (interfering with contact tracing and follow-up treatments); (3) healthcare factors: scabicide shortages and diversity, lack of examination privacy and staff inexperience; (4) organisational factors: overcrowding, ineffective interorganisational coordination, and lack of support and maltreatment by state authorities (eg, not providing basic facilities, obstruction of self-care by camp residents and non-governmental organisation (NGO) aid).

**Conclusions** We recommend development of accessible scabies guidelines for camps, use of consensus diagnostic criteria and oral ivermectin mass treatments. In addition, as much of the work described was by small, volunteer-

## STRENGTHS AND LIMITATIONS OF THIS STUDY

⇒ Recruitment through online healthcare staff networks enabled collection of the subjective perspectives of a participant population difficult to reach through formal means, given many had volunteered with non-governmental organisations but were not necessarily still anchored to them.

⇒ Telephone interviews prevented observation of non-verbal cues, but did facilitate international involvement in a timely and resource-efficient manner.

⇒ The relatively long and semistructured nature of the interviews gave participants opportunity to articulate their perspectives, to an extent unconstrained by researcher presumptions over what may have been important to them about their experiences.

⇒ Conducting data collection and thematic analysis simultaneously enabled assessment of data saturation during recruitment, but advertising and interviewing solely in English excluded experiences from non-English-speaking individuals.

⇒ The retrospective nature of the study may have introduced recall bias.

staffed NGOs, we in the wider healthcare community should reflect how to better support such initiatives and those they serve.

## INTRODUCTION

Scabies is a stigmatised contagious skin condition caused by infestation with the mite *Sarcoptes scabiei*.[1] Transmission is mainly skin-to-skin, less commonly via fomites such as bedding.[2] Evidence from epidemiology and experimental trials indicates scabies prevalence is probably not influenced by levels of personal body washing.[3] Symptoms begin 3–6 weeks after first infestation, as early as 1 day after reinfestation.[4] Secondary bacterial infections are common,[5 6] with potential for serious long-term health impacts, including chronic

rheumatic heart disease and chronic kidney disease.[7–9] Diagnosis is normally by clinical examination, sometimes using dermatoscopy.[4 10] However, definitive confirmation of diagnosis, according to consensus criteria of the International Alliance for Control of Scabies (IACS), requires 'A1: mites, eggs or faeces on light microscopy of skin samples; A2: mites, eggs or faeces visualised on individual using high-powered imaging device; A3: mite visualised on individual using dermoscopy' (see Box 1). Simultaneous

**Box 1    Summary of the 2018 International Alliance for Control of Scabies criteria for the diagnosis of scabies**

**A: Confirmed scabies**
At least one of:
A1: Mites, eggs or faeces on light microscopy of skin samples.
A2: Mites, eggs or faeces visualised on individual using high-powered imaging device.
A3: Mite visualised on individual using dermoscopy.
**B: Clinical scabies**
At least one of:
B1: Scabies burrows.
B2: Typical lesions affecting male genitalia.
B3: Typical lesions in a typical distribution and two history features.
**C: Suspected scabies**
One of:
C1: Typical lesions in a typical distribution and one history feature.
C2: Atypical lesions or atypical distribution and two history features.
**History features**
H1: Itch.
H2: Close contact with an individual who has itch or typical lesions in a typical distribution.
*Notes:*
1. These criteria should be used in conjunction with the full explanatory notes and definitions (in preparation).
2. Diagnosis can be made at one of the three levels (A, B or C).
3. A diagnosis of clinical and suspected scabies should only be made if other differential diagnoses are considered less likely than scabies.
Attribution: Reprinted from Engelman *et al*,[42] which is licensed under CC BY 4.0 (https://creativecommons.org/licenses/by/4.0). No changes have been made.

treatment for diagnosed individuals and close contacts is required.[11] This normally consists of topical scabicides applied to the entire body or oral ivermectin, in both cases often given two times (1 week apart).[12] Environmental decontamination advice varies.[12] The WHO designates scabies as a neglected tropical disease (NTD), with an estimated 455 million cases annually.[13] Its highest burden is in the tropics, but in lower-burdened regions (such as Europe), scabies remains common, emerging as a public health problem when institutional outbreaks occur.[10 13 14]

Over the last decade, conflicts triggered large numbers of people to migrate to Europe.[15–17] By the end of 2016, European countries hosted 5.2 million refugees, 2.9 million in Turkey alone (mostly from Syria).[15] As a result, during 2014–2017, many formal refugee/migrant camps and reception centres, and informal (often illegally occupied) camps came into being or expanded (figure 1). Healthcare services in camps were often limited to clinics operating from caravans or makeshift structures, largely staffed by volunteers; better resourced camps could still face shortages and overcrowding following unexpected influxes.[16 18] Scabies has been one of the most frequently observed medical conditions in these settings.[18–22] In Germany, it was the third most common outbreak type in asylum-seeker shelters.[21] In France, Doctors of the World UK estimated in 2015 that up to 40% of those seeking their care in the Calais 'Jungle' camp had scabies (Cooper, online supplemental file 1, p. 2). Outbreaks have been widely discussed in news and social media. Media analysis by Seebach *et al*[23] described some of this coverage as stigmatising those affected by scabies, while positing migrants/refugees more generally as an invasive health threat to European countries. Multiple camps in France have been evicted with the public justification of responding to scabies outbreaks (examples of newspaper reports: Lichfield and Newton, online supplemental file 1, p. 2). Despite all this, to our knowledge, no study prior to ours has been published specifically on scabies outbreak diagnosis, treatment and management in refugee/

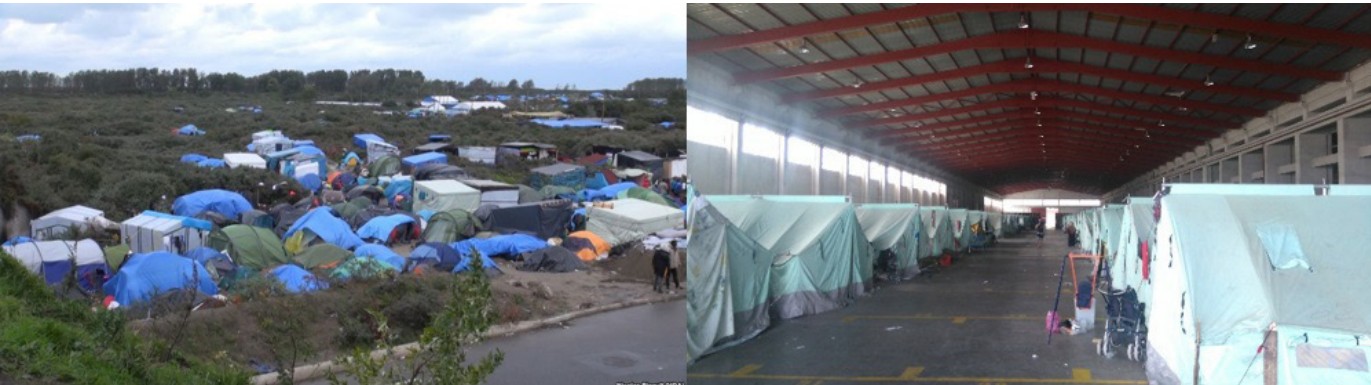

**Figure 1**    Informal and formal refugee/migrant camps. Left: informal camp in Calais, France, 2015 (VOA News-Nicolas Pinault (2015), public domain, original: https://web.archive.org/web/20210406154625/https://commons.wikimedia.org/wiki/File:Calais2015a.jpg). Right: formal camp in Thessaloniki, Greece, 2016 (Konstantinos Stampoulis (2016), used under CC BY-SA 4.0 which also applies to this figure, original: https://web.archive.org/web/20210406155208/https://commons.wikimedia.org/wiki/File:Refugee_hotspot_Fessas_(3).jpg).

migrant camps in Europe during this period. Further, we are unaware of any published study on healthcare staff perspectives on barriers and facilitators to treating and managing scabies in refugee camps globally.

## Aims

This qualitative study aimed to provide insights into experiences and perspectives of healthcare staff who treated scabies or managed associated outbreaks in refugee/migrant camps in Europe 2014–2017—specifically, to describe: (1) methods used to diagnose, treat and manage scabies; (2) camp characteristics and perceived barriers and facilitators to effective scabies outbreak management.

## METHODS
### Study design

We carried out a retrospective qualitative study using semistructured telephone interviews and framework analysis. Recruitment was through online networks of healthcare staff active in refugee/migrant settings. Reporting is in line with the Standards for Reporting Qualitative Research[24] (checklist, online supplemental table 1; and researcher characteristics, online supplemental file 1).

### Participant selection and recruitment

Healthcare staff from state and non-governmental organisations (NGOs) who had treated scabies in refugee/migrant camps in Europe, and/or managed associated outbreaks in the previous 3 years were eligible. We recruited by contacting:

▶ Relevant healthcare staff known to us.
▶ Individuals publishing research on medical care in refugee/migrant settings.
▶ Through online networks of healthcare staff active in refugee/migrant settings, including the alumni network of the European Centre for Disease Control fellowship programme and groups hosted on facebook.com. To target the latter, a public page (https://web.archive.org/web/20180523163917/https://www.facebook.com/scabiesresearchproject/) was shared to 10 relevant facebook.com hosted groups (online supplemental table 2) and messages sent to members whose posts indicated they fitted the inclusion criteria.

### Data collection, processing and analysis

Single audio-recorded, semistructured telephone interviews were conducted from a private room in the Department of Primary Care and Public Health at Brighton and Sussex Medical School by the first author NAR (who was alone) between November 2017 and February 2018. Participants gave data on personal characteristics and experience, and interviews followed a topic guide (online supplemental file 1) based on published literature and guidance from experts in scabies (epidemiology, JAC; dermatology, SLW; medical acarology, JM). They were scheduled for c. 40 min, but so as to provide sufficient information power,[25] participants were told they could speak longer as necessary, and to not feel precluded by the topic guide from relating any experiences and perspectives they thought relevant to the study aims. NAR conducted data collection and thematic analysis simultaneously to determine when data saturation was reached,[26 27] which was prospectively defined as being no new themes emerging in three consecutive interviews. Once data saturation was observed, no new interviews were conducted as the team could 'be reasonably assured that further data collection would yield similar results and serve to confirm emerging themes and conclusions.'[28] Field notes were not made during interviews, instead verbatim transcription was undertaken by professional service (9 of 12 interviews) or by NAR (3). NAR then coded text in NVivo V.11 (QSR International, Melbourne, Australia), generated an initial framework matrix and exported it to Microsoft Excel, in which qualitative framework analysis was completed (as outlined by Gale et al[29]). Groups of meaningful concepts were sought in the data and arranged hierarchically into main themes and subthemes. Each transcript was studied again twice for more evidence of the concepts. Subsequently, author JM reviewed the reported themes and subthemes and read the transcripts, finding no new relevant themes or subthemes than those grouped by NAR. Participants were not asked to subsequently comment on their transcripts or the general findings.

### Patient and public involvement

Patients and the public were not involved in the research design. Members of healthcare staff networks aided recruitment. CS is an aid worker with scabies outbreak experience in the study setting and coauthored the paper (they were not an interviewee).

## RESULTS
### Participants

All potential participants who responded to our advertising agreed to be interviewed after initial discussion of the study. Recruitment halted after 12 interviews as data saturation was assessed to have been reached (no new themes emerged in the last three interviews). Interviews lasted 34–71 min (mean 47). The 12 interviewees consisted of 4 doctors, 4 nurses, 3 allied health workers (AHWs: an ECG technician, podiatrist and a first aider) and 1 medical student. Ten were NGO volunteers, one an independent volunteer, one was employed by a public health service. Six had worked in informal camps, nine in formal ones, across camps at 15 locations within seven European countries: Greece, France, the Netherlands, Serbia, Belgium, Turkey and Macedonia. Geographical distribution of the camps (figure 2) reflected eastern entry migration routes into Europe, and clustering at the English Channel by those seeking to enter the UK. Experience in camps ranged from 4 days to 2.5 years (median

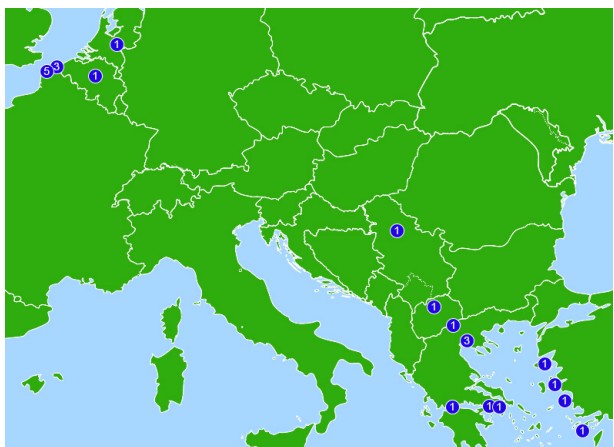

**Figure 2** Camp locations and number of participants (n=12) with experience at each site. Distribution of camps follows eastern entry migration routes into Europe, and clustering at the English Channel by those seeking to enter the UK. Left to right: Calais, France, 5 participants all active at informal (IF) camps; Dunkirk, France, 3 all IF; Brussels, Belgium, 1 formal (F); Heumensoord, the Netherlands, 1 F; Belgrade, Serbia, 1 IF; Macedonia, 1 IF; Patras, Greece, 1 IF; Idomeni, Greece, 1 IF; Thessaloniki, Greece, 3 (2 F, 2 IF); Athens, Greece, 1 IF; Attica, Greece, 1 F; Lesbos, Greece, 3 (1 F, 2 IF); Çeşme, Turkey, 1 IF; Samos, Greece, 2 F; Tilos, Greece, 1 F. Figure uses adapted basemap by Julio Reis (2006) created using public domain mapping from the European Environment Agency, used and changes made under CC BY-SA 3.0 which also applies to this image. Original basemap: https://web.archive.org/web/20161019140631/https://commons.wikimedia.org/wiki/File:Europe_biogeography_blank.svg.

84 days, IQR 233–251 days) (participant characteristics, online supplemental table 3).

### Diagnosis, treatment and outbreak management

Scabies diagnosis, treatment and outbreak management are summarised in table 1. Eleven interviewees (92%) diagnosed scabies based on history and clinical findings. Two AHWs and the medical student confirmed diagnoses with a doctor. One participant occasionally used skin scrapings in a Greek formal camp and during an outbreak in Serbia. In one camp, diagnosis was routinely given by a dermatologist and confirmed by microscopic examination. No one reported dermatoscopy. Three organisations attempted to follow Médecins Sans Frontiers' guidelines for ordinary scabies treatment[30] (online supplemental file 1), one combined this with British National Formulary advice.[31] Another had their own scabies protocol.

Only seven participants (58%) reported scabicides were provided in camps they worked in while they were there. The remaining five (42%) stated only symptomatic management was offered (detailed in table 1) because it was felt proper treatment would be futile due to high reinfestation risks (four participants) or due to prescribing restrictions (one participant). One-third of participants said treatments varied, depending on availability. When NGOs offered scabicides, they were generally given to anyone complaining of itch, due to a low threshold of suspicion (five of seven). However, one NGO only treated individuals with a dermatologist-confirmed diagnosis, another only treated 'serious cases'.

Eleven participants (92%) believed scabies was difficult to manage, whereas a general practitioner (GP) following a formal camp's scabies protocol considered it 'just a nuisance' (P7). Interviewees described high variation in outbreak management (table 1), particularly in timings used for permethrin application. These ranged from singular treatment to second applications up to a week later. Three participants (25%) could not recall timings or if close contacts were treated. In four interviews, participants claimed to treat all close contacts regardless of symptoms. Only three participants (25%) reported use of oral ivermectin for scabies, which was limited due to lack of availability. Five interviewees (42%) reported environmental decontamination was considered too challenging to conduct. Cleaning/replacing clothing and bedding was the most common form of decontamination. Individuals were given replacement clothing/bedding and old items washed (two of seven), disposed of (three of seven) or put into plastic bags for 72 hours (two of seven). One formal camp in Greece burned blankets. One organisation built an inventory of rental clothes for individuals while theirs were in bags. Generally, decontamination of furniture and shelters was not attempted (only mentioned in two interviews). Where it was, it involved disinfectant or covering furniture and insides of shelters in plastic sheeting for 72 hours.

### Camp characteristics and perceived barriers and facilitators to effective scabies outbreak management
#### Camp characteristics

All participants described camps as difficult places to work; legal status seemed unconnected to environmental quality. Participants described poor living standards (quotes, online supplemental file 1, pp. 9–11). For example, P8 described a formal camp (Greece):

> It's horrible. There's only one water source for over two thousand people, there's not enough clothes, people are sleeping outside, there's not enough tents, the hygiene is terrible. I was there in the summer so it was hot, there's not enough shading and people couldn't find a place outside the sun. It's more horrible now, as its winter, so it's freezing, and there's no heating. To be honest it's worse than the refugee camps I've seen in Africa.

P9, who had worked at informal and informal camps (France and Greece), spoke similarly:

> They are a human rights violation. I don't think people should be living in them, especially in a first world European country this should not be happening. People are living in cold tents without heating and electricity, there's rats, there's scabies, there's a lack of toileting facilities, a lack of hygiene, lack of food.

**Table 1** Diagnosis, treatment and outbreak management of scabies (n=12 participants)

| Diagnosis | | |
|---|---|---|
| Clinical features observed in patients | Itching, mainly at night | 12 |
| | Rash | 7 |
| | Burrow lines | 6 |
| | Infected lesions | 2 |
| Parts of the body examined | Hands | 8 |
| | Arms | 7 |
| | Torso | 6 |
| | Genitals | 4 |
| | Legs | 3 |
| | Feet | 3 |
| | Armpits | 2 |
| Equipment used during examination | Gloves | 7 |
| | Nothing | 2 |
| | Lights | 2 |
| | Microscope | 2 |
| | Dermatoscope | 0 |
| Duration of examination | <5 min | 8 |
| | 5–10 min | 2 |
| | >10 min | 1 |
| **Treatment** | | |
| Treatment guidelines | None | 8 |
| | MSF clinical guidance | 3 |
| | British National Formulary | 1 |
| | NGO's own protocol | 1 |
| Management of individual | Symptomatic management only | 5 |
| | Antihistamines | 3 |
| | Topical steroid | 1 |
| | Moisturiser | 1 |
| | Antibiotics for infections | 1 |
| | Varying treatment depending on availability | 4 |
| | Unknown topical treatment | 3 |
| | 1st line: permethrin 5% cream; 2nd line: benzyl benzoate; symptomatic treatment: antihistamines | 2 |
| | 1st line: ivermectin PO; 2nd line (or if ≤15 kg): permethrin 5% cream | 2 |
| | Permethrin 5% cream only | 1 |
| Who was given acaricide treatment | No one | 5 |
| | Anyone with itching, plus close contacts | 4 |
| | Anyone with itching | 1 |
| | Only 'serious cases' | 1 |
| | Cases confirmed by dermatologist, plus close contacts | 1 |
| **Outbreak management** | | |
| Timing of mass treatments | No treatment | 4 |
| | Unknown | 3 |
| | Cases treated twice, 1 week apart (permethrin) | 1 |
| | Cases and symptomatic contacts treated ×2 a few days apart (permethrin), asymptomatic contacts ×1 | 1 |
| | Cases treated ×1, 3 days apart (permethrin) | 1 |
| | Cases treated ×2, 24 hours apart (permethrin and benzyl benzoate) | 1 |
| | Case treated ×1 (permethrin) | 1 |
| | Cases treated ×1 (ivermectin) | 1 |

**Table 1** Continued

| Diagnosis | | | |
|---|---|---|---|
| Treatment of close contacts | No treatment available to anyone | | 4 |
| | All close contacts | | 4 |
| | Unknown | | 3 |
| | Symptomatic contacts | | 1 |
| Environmental decontamination | Clothing | | 7 |
| | Bedding | | 7 |
| | None | | 5 |
| | Furniture | | 1 |
| | Shelters | | 2 |

MSF, Médecins Sans Frontiers; NGO, non-governmental organisation; PO, orally.

Ten participants (83%) believed shelter was inadequate or non-existent, two (17%) very basic. Twelve (100%) described lack of suitable water, sanitation and hygiene (WaSH) facilities. Five (42%) described safety worries regarding: fights between residents, police mistreatment and instability of camp structures.

**Barriers and facilitators to effective scabies outbreak management**

Themes and subthemes are illustrated by example quotes in table 2 (barriers) and table 3 (facilitators), with 16 pages of themed quotes in online supplemental file 1. Each interviewee described many barriers to effective scabies management; four key themes arose.

*Lack of WaSH facilities*: lack of access to washing facilities for fabrics to prevent reinfection was identified by 10 interviewees (83%). Three interviewees (25%) believed sharing belongings was promoting reinfestation, amplified by inability to wash them between users. Six interviewees (50%) described difficulties associated with camp residents in ability to wash themselves, due to non-existent or insufficient showers. This was especially relevant regarding topical scabicides, which should be washed off.[4]

*Social barriers*: eight interviewees (67%) felt language differences obstructed understanding individuals' problems and explaining treatment and decontamination. Four (33%) felt stigma associated with scabies adversely affected management, as it reduced presentation to services and individuals were less likely to inform close contacts who required treatment. Treatment non-compliance was considered a barrier in four interviews (33%), caused by complicated management regimes and residents' competing priorities. Population mobility concerned six interviewees (50%), who felt it affected contact tracing, follow-up treatment and disrupted elimination efforts.

*Organisational barriers*: six participants (50%) perceived overcrowding as a major barrier to effective management. Large dormitory-style rooms were said to be the hardest in which to determine close contacts, as were bigger units in which blanket partitions were used between families. Five interviewees (42%) described poor coordination between organisations (both NGOs and governmental organisations) as a barrier. This primarily involved organisations providing different resources not communicating about appropriate timings (eg, lack of connection between treatments, clothing replacement, access to showers). Two participants described disrespectful care and stigmatisation by healthcare workers. Ten participants (83%) described how lack of support and mistreatment by state authorities acted as a major barrier, most of which was described in informal camp settings (7 of 10). Two interviewees mentioned residents facing police brutality, including having tents slashed, and clothing, shoes and bedding confiscated. This encouraged sharing (with potential implications for transmission), and by adding further challenges to everyday life, it reduced volunteer and patient capacity to focus on treatment and decontamination. One participant reported authorities 'sometimes they cut… water, they try not to get us to be able to give out clothes, they raided our distribution place' (P2). Their explanation was: 'What the government is trying to do is, they're trying not to get the refugees too comfortable.'

*Healthcare barriers*: eight interviewees (67%) considered scabicide availability a major barrier. Cost, prescribing restrictions and variations in pharmacy stocking contributed; for most NGOs, availability relied on donations. Inconsistency complicated treatment and fostered mistrust. Three interviewees (25%) felt lack of privacy impacted care. Some residents attempted to circumvent this by providing photos on phones, but this was not considered as useful as examinations. Five interviewees (42%) felt healthcare workers accustomed to European work were unfamiliar with scabies, which was exacerbated by volunteer turnover.

*Facilitators*: two participants (17%) said they did not experience factors aiding scabies treatment and management, the remaining described five, when available: WaSH facilities, language facilitators, education, treatment incentives and coordination of organisations (table 3). Three (25%) felt access to showers and washing machines was vital, as clothing and bedding could be reused without concern

**Table 2** Barriers to providing effective scabies management

| Key themes *Subthemes* | Comments from participants |
|---|---|
| **Lack of WaSH facilities** | |
| *Inability to wash belongings* | 'Part of the treatment is also to wash the bedding, and all clothing in hot water, dry it in a drier… those facilities weren't readily available for anybody at the camp… even if we'd had a sort of endless supply of Permethrin or Ivermectin, I think it still would have been problematic with this problem of re-infection.' (P6) |
| *Lack of shower facilities* | 'There is really no shower and there are some areas that are a kind of shower area, but it's not really a shower and its cold water.' (P5) 'Treating scabies was problematic because… the lotion, you're supposed to keep it on for 12 hours and then wash it off… and access to showers was not always that easy.' (P6) |
| *Sharing of belongings* | 'Charities came in to give clothes, but the clothes weren't ever washed and when someone went across the border… their stuff was left, and whoever was in the tent just used that.' (P1) |
| **Social** | |
| *Language barriers* | 'They don't understand why they have to use the cream and what's going on. So, they feel reluctant to comply, and they're like no, it doesn't work. If you don't have proper translators to translate anything to you that also could be a big barrier.' (P2) |
| *Stigma* | 'Didn't want to tell the other people they had scabies… there is some sort of stigma of course if you have some sort of rash that you are not clean or something.' (P8) |
| *Non-compliance with treatment* | 'I would say to put the stuff all over your body, head down to the toes… to do it all in one night and keep it on for so many hours and then they walk away and then only put a bit of cream here, and a bit of cream there, for a week and then they come back and it hasn't got better.' (P7) |
| *Transient population* | 'You have to trace who might be the contact… in the refugee setting you have to also to treat patients that might be from the same family, and they just move around. You can't really tell who has slept here, who has not slept here, who might possibly get it.' (P2) |
| **Healthcare** | |
| *Availability of medication* | 'We don't have the treatment for it, because we don't have the money to buy in the treatment.' (P4) 'The medication available is the least convenient medication you could use. If you could use a pill for everyone that would be so much easier.' (P10) |
| *Lack of privacy for medical examinations* | 'Everything sees what we are doing, so if someone has to show something we don't have a closed examination room… you can't really examine them… you can't really touch and see in your own eyes… you have to rely on the photos because they're too shy to show it.' (P2) |
| *Inexperienced staff* | 'European doctors did not have a lot of experience with scabies, so I think a lot of them didn't realize what they were seeing.' (P10) |
| **Organisational** | |
| *Crowded living conditions* | 'People were living in fairly cramped conditions which obviously made transmission a high possibility.' (P6) |
| *Poor coordination between organisations* | 'Lack of coordination and appropriate communication… I don't think it's the resources; the resources are there they're just not put together in the right way.' (P10) 'Logistical problems, sufficient manpower, sufficient washing machines, sufficient bedding, who cares for this?' (P11) |
| *Poor treatment by authorities* | 'Refugees have their bedding and their coats and shoes confiscated by police at night.' (P3) 'People were hiding from the authorities… the settlement or their place where they would sleep would not be there because tents were being like routinely slashed and belongings were being destroyed by the police and authorities.' (P12) |

Further themed participant quotes in online supplemental file 1, pp. 11–21.
WaSH, water, sanitation and hygiene.

of reinfection, and topical treatment guidelines adhered to. Nine interviewees (75%) described the importance of translators in overcoming language barriers. Only three NGOs used official translators, the rest were aided by residents. One participant described using recorded voice messages and information sheets in different languages. Four (33%) believed resident education aided outbreak management, by alerting them to others who needed treatment and by providing education on correct treatment application. This was also thought to have reduced stigma within camps. Four (33%) stated incentives (eg, new clothing or just underwear) aided treatment compliance. Eight interviewees (67%) reported positive impacts from interorganisational communication, including information and resource sharing, and enabling referrals. The GP who described scabies as 'just a nuisance' (P7) attributed that to having a protocol followed by all organisations. In particular, the 'ability to pass on a systematic and well-rehearsed message' (P7) was thought to have improved adherence.

## DISCUSSION
### Principal findings
Participants reported in the camps they had worked, scabies diagnosis was primarily clinical (without dermatoscopy), and treatment and outbreak management varied highly. Nearly half said scabicides were unavailable; only

**Table 3** Facilitators to providing effective scabies management

| Key themes *Subthemes* | Comments from participants |
| --- | --- |
| WaSH facilities | 'People have access to showers… clothes had to be washed in the washing machines… everything else was taken out and burnt. That was the protocol… we didn't have outbreaks.' (P7) |
| Social | |
| Language facilitators | 'We had our own translator who spoke several languages, and there were people who lived in the camp, working or volunteering as translators.' (P1) |
| Education | 'If you educate the patient very well, make them understand very well what's going on, and what needs to be done… They understand, and they start to even manage to recognise their friends.' (P2) |
| Treatment incentive | 'When they found out we would give them another set of clothes and we would buy underwear for them, some more people were interested as they had an incentive to take part.' (P8) |
| Coordination of organisations | 'We give them a cream 24 hours before and then asked them to get showers, and then we put the creams again. So we partner with shower teams, and also, we partner with people from the distribution team so we could get them new fresh clothes.' (P2) |

Further themed participant quotes in online supplemental file 1, pp. 21–24.
WaSH, water, sanitation and hygiene.

symptomatic management was offered. They described camps as difficult places to work, with poor living standards for residents. Key perceived barriers to control were (1) lack of WaSH facilities, specifically: absent/limited showers (difficult to wash off topical scabicides), and inability to wash clothes and bedding (may have increased transmission/reinfestation); (2) social factors: language, stigma, treatment non-compliance and mobility (interfering with contact tracing and follow-up treatments); (3) healthcare factors: scabicide shortages and diversity, lack of examination privacy and staff inexperience; (4) organisational factors: overcrowding, ineffective inter-organisational coordination, and lack of support and maltreatment by state authorities (eg, not providing basic facilities, obstruction of self-care by camp residents and NGO aid).

### Strengths and weaknesses

This qualitative study succeeded in giving voice to the subjective perspectives and experiences of healthcare staff involved in this previously unstudied topic. However, recruitment methods may have prevented fully representative participant selection. First, because advertising, recruitment and interviews were conducted solely in English, the experiences of non-English-speaking individuals were excluded. Second, as we assessed data saturation had been reached and halted recruitment, there was a fairly small number of participants. However, while 12 participants might be small for a quantitative study, our sample size does correspond with what Guest et al[26] determined was usually sufficient to reach data saturation in qualitative studies focused on narrow topics. Seeking to 'make evidence-based recommendations regarding non-probabilistic sample sizes for interviews',[26] they found saturation not only usually occurred in the first 12 interviews, but that key constituents of meta-themes presented as early as in the first 6. One study strength was that the relatively long semistructured interviews gave participants opportunity to articulate their perspectives relatively unconstrained by researcher presumptions over what may have been important to them about their experiences. The resultant transcripts provided a large amount of rich data on the perspectives and experiences of the participants (for example, see extensive quotes in online supplemental file 1, pp. 9–24), and thus had far more 'information power'[25] than could be expected from a short survey with higher participant numbers. Telephone interviews prevented observation of nonverbal cues, but did enable international recruitment in a timely and resource-efficient manner. However, the retrospective nature of the study may have introduced recall bias. Future work in refugee/migrant camps could reduce this limitation by interviewing healthcare staff during outbreaks, either remotely or in person. Our study aimed only to describe the perspectives and experiences of the participants, seeking to understand the subjective meaning of the phenomena to them. Given these aims, as our study draws on the understanding of the research subjects themselves, its method is high on internal validity, but the findings are not necessarily generalisable. We take the latter limitation as a given, but not one that invalidates the study. However, inference and public health recommendations based on this research must be pragmatic with limitations acknowledged.

Louka et al[22] report scabies epidemiology in refugee/migrant populations in Greece, while more general quantitative studies from across Europe touch on scabies in these populations (for example, see [19–21]). However, to our knowledge, our study is the only qualitative work published on healthcare staff perspectives on barriers and facilitators to treating and managing scabies within refugee/migrant camps, either in Europe or globally. This underlines the major strength of our study in providing a platform for these under-represented healthcare voices, yet given this it is difficult to compare our findings with others. Wollina et al[32] do usefully recount their related experience in a German hospital which received dermatological referrals (scabies being the most common) from a nearby formal refugee camp. Even in

this more resourced setting, language barriers were an issue, as they were for some of our participants. An environmental health assessment of a French informal camp with a high scabies incidence detailed a very similar environment to those described by many of our participants: 'Shelters were vastly inadequate and directly exacerbating the ill-health and psychological distress suffered by some residents of the camp… Residents at all sites reported difficulties with washing themselves, their clothes and their bedding.'[18]

### The wider context of scabies across Europe

Some studies suggest scabies infestations may have increased across Europe recently, including research conducted in Turkey, France and the Netherlands, countries in which a number of our study participants were based.[33–38] In the general population of the Netherlands, recorded scabies diagnosis per 1000 persons per year increased fourfold in 2011–2020 (0.6–2.6), in parallel with a sixfold increase in scabicide dispensing over the same period. Most dispensations were permethrin; however, large nosocomial outbreaks were linked to a peak of ivermectin prescriptions in 2014–2015.[34] Similarly, German studies found increasing numbers of cases since 2009, most notably an 11-fold increase in persons 15–24 years of age from 2009 to 2018.[35 36 38] In Croatia, where reporting of scabies is mandatory, general incidence increased sixfold in 2007–2017, and multiple outbreaks were reported from adult care and nursing homes.[37] Some data suggest an association between population movement and higher incidence in the general population. However, though refugee/migrant groups have been disproportionately affected by scabies when living in the crowded camps which are the subject of our study, they are unlikely to be the cause of changes in reported overall prevalence within the wider population.[33 35–38] Suggested explanations given for this claimed trend include increased number of sexual contacts, treatment failure in younger patients with presumed poorer compliance and ageing populations.[33 34 36–40] The observed trend may also be in part a result of reporting bias, given the higher attention scabies is now being given in some countries following its adoption as a WHO NTD.

### Implications for clinicians and policymakers
#### Diagnosis and treatment
It is unsurprising no participants reported dermatoscopy use given equipment expense, paucity of trained individuals and its reduced sensitivity when outbreak size limits examination time.[41] However, subsequently published (2018) consensus criteria for scabies diagnosis developed by IACS (https://www.controlscabies.org) provides a standard for clinical diagnosis. This is summarised in Box 1 and fully detailed in Engelman et al,[42] though it requires validating in refugee/migrant camp settings. Some participants suggested screening on entry to camps to reduce ingress of new infestations. This was also suggested by Wollina et al,[32] has been carried out in reception centres[43]

and may be feasible in camps. However, in informal camps, screening may not be consented to by those seeking entry and NGOs may have insufficient resources to adequately screen, potentially leading to false positives/negatives. The asymptomatic incubation period would also mean some potential outbreak index cases would still be missed.

Participants reported problems using topical scabicides. First, topical treatment involves full body coverage with creams, left on for at least half a day. Privacy was limited and individuals living in close proximity may have wished to avoid being seen treating themselves for a stigmatised condition, thus discouraging appropriate application. Second, topical scabicides need to be washed off to reduce potential irritancy,[4] but facilities were usually only minimally available (if at all). Oral ivermectin mass drug administrations (MDAs) are easier to administer, and were conducted successfully in Dutch and German reception centres during the period.[43 44] However, ivermectin is expensive in Europe and unlicensed for scabies in most European countries[12] despite a good safety profile[45 46] and inclusion in the WHO's essential medicines list.[47] The potential health impacts of chronic conditions secondary to scabies[7–9] support the value of timely MDAs in these settings. Participants reported widespread use of camp residents as informal translators, but this raises confidentiality concerns, and ad-hoc interpreters often misinterpret/omit important medical information.[48]

### Outbreak management and camp environments
Discrepancies in management reflect lack of widely accepted standard outbreak guidelines. Guidance (for example, 30 31) is often for individual households and does not make adaptions based on the logistical difficulties of camp environments. Institutional scabies outbreak guidelines exist, yet none are based on a systematic review of the published evidence.[12] This is also the case for the 'European Guideline'.[49] More broadly, there is a lack of research regarding the most effective interventions for preventing scabies transmission to close contacts,[50] particularly in refugee/migrant settings. Organisations often favoured attempting to decontaminate clothing and bedding rather than shelters, yet living mites have been found in dust samples on floors and furniture.[51] Variance in reported decontamination reflects a sparse historical evidence base.[12] However, recent experimental work[52] could support uniform action. Where washing/drying machines are available (few camps in this study), ≥10 min at ≥50°C will kill all *S. scabiei* mites and eggs.[52] Lacking such resources, some participants reported isolating potential fomites with plastic bags/sheeting for 72 hours. Due to mite desiccation, this may have been effective in temperate-dry settings, but if bags were left outdoors in the cold and damp, mite survival can be longer. At the outer limit, in warm-humid conditions, elimination can take >8 days.[52] Existing guidance should be updated. Even though improving WaSH facilities is unlikely to reduce skin-to-skin transmission,[3] it nevertheless would support treatment and control where transmission via

fomites may be a risk, when topical scabicides are used and to minimise secondary infections.[14] It could also be expected to reduce transmission of other diseases. It should however be noted that (primarily) ivermectin-based MDAs have achieved very good results in community intervention trials across Oceania (for a summary, see Middleton[53]) without any environmental measures on hygiene, disinfection of bedding, etc (for example, see work in the Solomon Islands[54]). Participants described crowded and substandard living conditions that likely provided an ideal environment for scabies transmission, while obstructing treatment and outbreak management. Some believed lack of political will was preventing provision of suitable living conditions and access to healthcare, as governments feared they would encourage people to stay too long. Going further, participants reported actual obstruction by authorities of resident self-care and NGO aid efforts, with implications for transmission and control of scabies.

### The meaning of common scabies occurrence in camps in Europe

Scabies prevalence differs between world regions, but its occurrence is near ubiquitous in human populations worldwide.[4 13] It follows from this premise that ingress of *S. scabiei* into at least some of the camps was practically inevitable. However, the observed regularity and size of outbreaks was not. We aimed to provide insights into experiences and perspectives of healthcare staff who treated scabies or managed associated outbreaks in refugee/migrant camps in Europe 2014–2017, but we did not set out to specifically determine the meaning, the explanation, underlying why they took place. To our knowledge, the only published research with an objective close to that is Louka *et al*,[22] who aimed to 'investigate changes over time [in scabies epidemiology in health care centres for refugees and asylum seekers in Greece] and factors relating to these changes'. They note that 'outbreaks of scabies cases coincided with peaks in other infectious diseases' and suggest a possible explanation may be 'an increase of the refugees/asylum seekers residing in the centers at the specific time points, [and] the crowded conditions within the centers'.[22] This suggestion is in line with prior studies which demonstrated that high density of potential hosts is a major transmission driver of scabies.[14] For a detailed overview of scabies transmission drivers in other semiclosed institutional settings where outbreaks are a regular occurrence (eg, residential settings for elderly people, children and those with learning disabilities; prisons; schools; hospitals and hostels), see Middleton *et al*.[14] Many of these drivers were mentioned by our participants in some form while discussing their experiences of diagnosis, treatment, outbreak management and camp environments. Specifically of those discussed by Middleton *et al*,[14] the following were mentioned by our study participants: high densities of potential hosts; residents moving between semiclosed units; communication difficulties; reduced access to appropriate treatment; diagnostic error and/or delay; reduced access to laundry.

It is beyond the capacity of this study to determine the relative contribution of these factors to outbreak regularity and size, and indeed their contribution is likely to have been different in different camps and at different times. Clinicians and policymakers tasked with managing scabies in similar refugee settings may find it helpful to carry out site-specific assessment of scabies transmission drivers to guide intervention and treatment, using the guidance on how to do so provided by Middleton *et al*.[14]

More broadly, our view is that without support and coordination of all those responsible for care of refugees/migrants, successful scabies management will remain challenging, and the resource poor and overcrowded aspects of camp environments which enable outbreaks will remain unchanged. Indeed, in late 2020, a similar situation to those described in 2014–2017 was observed by coauthor CS in a formal camp on Lesvos (Greece). The NGO CADUS, in which CS works, had been deployed within a WHO initiative to provide medical care. Scabies was widespread among the camp population of 7500, but the eradication programme had to begin with a capacity of 10 treatments per day. A small volunteer-staffed NGO provided treatment, including provision of (camp-external) showers, clothes and blankets exchange. The overcrowded camp lacked adequate facilities to wash clothes, and mostly only symptomatic treatment could be offered, which CADUS medical staff viewed as highly unsatisfactory.

### Future recommendations

Our study demonstrates the importance of having clear protocols for scabies diagnosis, treatment and management in refugee/migrant camps. Systematic evidence evaluation must be combined with input from individuals and organisations on the ground to provide high-quality feasible guidelines that explicitly consider camp settings, formal and informal. Given most participants were not doctors, guidelines need to be understandable to everyone involved. Though topicals should be provided for children and pregnant women, difficulties associated with them in camps could be avoided for most adult patients through wider licensing and availability of oral ivermectin. The 'WHO Informal Consultation on a Framework for Scabies Control' recommended populations with ≥10% prevalence should receive ivermectin MDAs[55 56]; we suggest this should include refugee/migrant camps. Strategies for lower levels of endemicity in camps should also be investigated. More research and advocacy for those living in refugee/migrant camps in Europe could help improve management of common health problems. This is particularly relevant for managing scabies in camps, where substandard conditions, inadequate resources and lack of guidance can lead to poor quality of care and ineffective treatment. As our participants indicate, much of this work is shouldered by small, volunteer-staffed NGOs operating

with minimal resources in highly challenging (often illegally occupied) settings. We in the wider healthcare community should reflect on how we can better support these initiatives, and those they serve.

## Author affiliations
¹Department of Primary Care and Public Health, Brighton and Sussex Medical School, Watson Building, University of Brighton, Falmer, UK
²Clinical Informatics Research Unit, Faculty of Medicine, University of Southampton, Southampton, UK
³CADUS, Berlin, Germany
⁴Faculty of Infectious and Tropical Diseases, London School of Hygiene and Tropical Medicine, London, UK
⁵Hospital for Tropical Diseases and Department of Dermatology, University College London Hospitals NHS Foundation Trust, London, UK
⁶NIHR Global Health Research Unit on Neglected Tropical Diseases, and NIHR Applied Research Collaboration Kent, Surrey and Sussex, Brighton and Sussex Medical School, Falmer, UK

**Acknowledgments** We thank the interviewees for their time and openness with our study, and the administrators and members of the healthcare staff networks who advertised our study online. We are also grateful to our colleagues at Brighton and Sussex Medical School: Elizabeth Ford (for input into study design), Jessica Stockdale (for producing figure 2) and Maya Khan (for setting up our Facebook recruitment pages).

**Contributors** This study was carried out as part of the work of the Scabies Research Team based at Brighton and Sussex Medical School, London School of Hygiene and Tropical Medicine, and University of Southampton. Authorship order is alphabetical by surname, except the first (primary researcher) and last (lead supervisor). For clarity, we detail contributions using the CRediT Contributor Taxonomy (https://credit.niso.org/), and provide employment and disciplinary descriptions. Conceptualisation—JM and JAC. Formal analysis—NAR and JM. Investigation—NAR and JM. Methodology—JM. Project administration—NAR and JM. Supervision—SL and JM. Writing (original draft)—NAR and JM. Visualisation—NAR and JM. Writing (review and editing)—NAR, JAC, MGH, SL, CS, SLW and JM. JM as guarantor accepts full responsibility for the finished work and/or the conduct of the study, had access to the data, and controlled the decision to publish. NAR is a global emergency medicine clinical fellow; JAC is a public health physician and epidemiologist; MGH is a senior research fellow (global health); SL is a research coordinator; CS is an aid worker; SLW is a consultant dermatologist and associate professor (infectious and tropical diseases); JM is a research fellow (public health; neglected tropical skin diseases).

**Funding** This research was financially supported by Brighton and Sussex Medical School from internal funds (N/A). JM is funded by the National Institute for Health and Care Research (NIHR) Applied Research Collaboration Kent, Surrey, Sussex (NIHR200179) and the NIHR Global Health Research Unit on Neglected Tropical Diseases (NIHR131996).

**Disclaimer** The views expressed are those of the authors and not necessarily those of the NHS, the NIHR or the Department of Health and Social Care.

**Map disclaimer** The inclusion of any map (including the depiction of any boundaries therein), or of any geographic or locational reference, does not imply the expression of any opinion whatsoever on the part of BMJ concerning the legal status of any country, territory, jurisdiction or area or of its authorities. Any such expression remains solely that of the relevant source and is not endorsed by BMJ. Maps are provided without any warranty of any kind, either express or implied.

**Competing interests** None declared.

**Patient and public involvement** Patients and/or the public were involved in the design, or conduct, or reporting, or dissemination plans of this research. Refer to the Methods section for further details.

**Patient consent for publication** Not required.

**Ethics** This study involves human participants and was approved by the Brighton and Sussex Medical School Research Governance and Ethics Committee (ER/BSMS3398/1).

**Provenance and peer review** Not commissioned; externally peer reviewed.

**Data availability statement** Data are available upon reasonable request. All data relevant to the study are included in the article or uploaded as supplemental information. The online supplemental file includes 16 pages of themed quotes from interviews. Other data are available on reasonable request from the corresponding author.

**ORCID iDs**
Naomi A Richardson http://orcid.org/0000-0003-1151-6435
Jackie A Cassell http://orcid.org/0000-0003-0777-0385
Michael G Head http://orcid.org/0000-0003-1189-0531
Stephen L Walker http://orcid.org/0000-0002-2034-8376
Jo Middleton http://orcid.org/0000-0001-5951-6608

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
