## [Reviewer comments · BMJ Open]

ARTICLE DETAILS

TITLE (PROVISIONAL)	Scabies outbreak management in refugee/migrant camps in Europe 2014-17: a retrospective qualitative interview study of healthcare staff experiences and perspectives
AUTHORS	Richardson, Naomi; Cassell, Jackie; Head, Michael; Lanza, Stefania; Schaefer, Corinna; Walker, Stephen; Middleton, Jo

VERSION 1 – REVIEW

REVIEWER	Lugović-Mihić, Liborija University of Zagreb
REVIEW RETURNED	23-May-2023

GENERAL COMMENTS	The manuscript "Scabies outbreak management in refugee/migrant camps in Europe 2014-17: a retrospective qualitative interview study of healthcare staff experiences and perspectives" is a valuable and extensive manuscript with complex statistical analysis, which presents important factors related with scabies occurrence in refugee/migrant camps in Europe. Thus, it includes a complex analysis, with visual schematic presentation, geographich presentation of obtained experiences from few countries. Namely, in their research, authors conducted complex retrospective qualitative study using semi-structured telephone interviews and framework analysis (recruitment primarily through online networks of healthcare workers in refugee/migrant settings), based on data from refugee/migrant camps in Europe 2014–17. The study included 20 participants (4 doctors, 4 nurses, 3 allied health workers, 1 medical student) who had worked in camps across 15 locations within 7 European countries (Greece, Serbia, Macedonia, Turkey, France, Netherlands, Belgium). According to their results, participants reported that in camps they had worked, scabies diagnosis was primarily clinical (without dermatoscopy), and treatment and outbreak management varied highly (7 stated scabicides were provided, 5 that only symptomatic management was offered). They described camps as difficult places to work, with poor living standards for residents. Key perceived barriers to scabies control were lack of water, sanitation and hygiene (e.g. absent/limited showers); social factors (language; stigma; treatment non-compliance; mobility); healthcare factors (scabicide shortages and diversity); lack of examination privacy; staff inexperience; organisational factors (overcrowding; ineffective interorganizational coordination); lack of support and maltreatment by state authorities. However, there are some suggestions for the authors. Some parts of the text: - It is written: „Definite diagnosis has been classified as requiring ‘clinical features suggestive of scabies... plus a visible mite under dermatoscopy... or a skin scraping or biopsy with identified mites,
---

	75 mite eggs, or faeces“ – It is necessary to add/emphasize that correct definite (precise) diagnosis of scabies is made via mineral oil examination with the microscopic identification of mites or their eggs or feces (scybala)(confirmation by scraping and identification of the mite is crucial). -The authors could focus more on the meaning of their results (and other similar study results), with a deep review of previous literature/studies data in this exact field. Since in the actual manuscript there is not even a separate chapter/section on this topic (the meaning of common scabies occurrence in camps in Europe), I recommend adding it. It would be also welcome to provide more information about experiences on scabies occurrence and therapy from those/mentioned countries (based on research and articles that talk about the frequency of scabies in them), or in neighboring countries, or countries from that area. Although there is general awareness of this problem (e.g. low hygiene and frequent scabies in camps), the same factor can be connected with the situation in specific countries and the increase in the frequency of scabies.
--	---

REVIEWER	Thornley, Simon The University of Auckland, Section of Epidemiology and Biostatistics
REVIEW RETURNED	29-May-2023

GENERAL COMMENTS	This is an important article which raises the issue of an ad hoc approach to a common problem of scabies in refugee camps in Europe. I think it raises the issue of guidelines for standardised diagnosis and treatment. The only issue I think should be improved is perhaps drawing more attention to the IACS guidelines. Also, in the Pacific, very good results of ivermectin MDA have been achieved without any environmental measures on hygiene or disinfection of bedding etc. This should also be pointed out.
---

VERSION 1 – AUTHOR RESPONSE

REVIEWER 1

RESPONSE: We are grateful for your positive evaluation of our study and your description of it as “a valuable and extensive manuscript with... a complex analysis, with visual schematic presentation, of obtained experiences from few countries. Namely, in their research, authors conducted complex retrospective qualitative study using semi-structured telephone interviews and framework analysis (recruitment primarily through online networks of healthcare workers in refugee/migrant settings), based on data from refugee/migrant camps in Europe 2014–17.” Thank you.

RI: “However, there are some suggestions for the authors.”

RESPONSE: We list below, point-by-point, the changes we have made in response.

R1: “It is written: “Definite diagnosis has been classified as requiring ‘clinical features suggestive of scabies... plus a visible mite under dermatoscopy... or a skin scraping or biopsy with identified mites, 75 mite eggs, or faeces“ – It is necessary to add/emphasize that correct definite (precise) diagnosis of scabies is made via mineral oil examination with the microscopic identification of mites or their eggs or feces (scybala)(confirmation by scraping and identification of the mite is crucial)..”

RESPONSE: We are thankful for this suggestion and have removed the previous text and instead added in text from the 2018 consensus criteria of the International Alliance for Control of Scabies on definite (i.e., confirmed) scabies diagnosis, including re microscopy. We have also added the full

summary of 2018 IACS criteria for the diagnosis of scabies as a Box, which can be found at the end of the main article (see end of our comments to reviewer 2 for full text of the box). It now reads (75-82 in tracked version, inserted text in [square brackets], deleted text strikethrough):

'Diagnosis is normally by clinical examination, sometimes using dermatoscopy.[4, [10]] [However, d]Definitive [confirmation of] diagnosis[, according to consensus criteria of the International Alliance for Control of Scabies (IACS),] has been classified as require[es]ing 'clinical features suggestive of scabies... plus a visible mite under dermatoscopy... or a skin scraping or biopsy with identified ['A1: M]mites, miteeggs, or faeces [on light microscopy of skin samples; A2: Mites, eggs or feces visualized on individual using high-powered imaging device; A3: Mite visualized on individual using dermoscopy' (see Box 1, Summary of 2018 IACS criteria for the diagnosis of scabies)].[10]

R1: "The authors could focus more on the meaning of their results (and other similar study results), with a deep review of previous literature/studies data in this exact field. Since in the actual manuscript there is not even a separate chapter/section on this topic (the meaning of common scabies occurrence in camps in Europe), I recommend adding it."

RESPONSE: Unfortunately, literature on this exact topic is very scarce. We already reference the key literature in our introduction section, but have followed your recommendation and added a new section to discuss 'the meaning of common scabies occurrence in camps in Europe', which we interpret as why outbreaks were occurring. We outline the related conclusion from the only study we know to have investigated the question directly, and contextualise it with what is known about scabies transmission drivers generally, and which ones were mentioned by our participants. The following new section is in discussion (414-440 in tracked version, inserted text in [square brackets]):

'[The meaning of common scabies occurrence in camps in Europe

Scabies prevalence differs between world regions, but its occurrence is near ubiquitous in human populations worldwide.[4,13] It follows from this premise that ingress of *S.scabiei* into at least some of the camps was practically inevitable. However, the observed regularity and size of outbreaks wasn't. We aimed to provide insights into experiences and perspectives of healthcare staff who treated scabies or managed associated outbreaks in refugee/migrant camps in Europe 2014–17, but we did not set out to specifically determine the meaning, the explanation, underlying why they took place. To our knowledge the only published research with an objective close to that is Louka et al.[22], who aimed to 'investigate changes over time [in scabies epidemiology in health care centres for refugees and asylum seekers in Greece] and factors relating to these changes'. They note that 'outbreaks of scabies cases coincided with peaks in other infectious diseases' and suggest a possible explanation may be 'an increase of the refugees/asylum seekers residing in the centers at the specific time points, [and] the crowded conditions within the centers'.[22] This suggestion is in line with prior studies which demonstrated that high density of potential hosts is a major transmission driver of scabies.[14] For a detailed overview of scabies transmission drivers in other semi-closed institutional settings where outbreaks are a regular occurrence (e.g., residential settings for elderly people, children and those with learning disabilities; prisons; schools; hospitals and hostels) see Middleton et al.[14] Many of these drivers were mentioned by our participants in some form whilst discussing their experiences of diagnosis, treatment, outbreak management, and camp environments. Specifically of those discussed in [14] the following were mentioned by our studies participants: high densities of potential hosts; residents moving between semi-closed units; communication difficulties; reduced access to appropriate treatment; diagnostic error and/or delay; reduced access to laundry. It is beyond the capacity of this study to determine the relative contribution of these factors to outbreak regularity and size, and indeed their contribution is likely to have been different in different camps, and at different times. Clinicians and policymakers tasked with managing scabies in similar refugee settings may find it helpful to carry out site-specific assessment of scabies transmission drivers to guide intervention and treatment, using the guidance on how to do so provided by Middleton et al.[14]]'

This is then followed by the summary conclusion, update, and future recommendations, which also allude to this question, though more obliquely.

R1: "It would be also welcome to provide more information about experiences on scabies occurrence and therapy from those/mentioned countries (based on research and articles that talk about the frequency of scabies in them), or in neighbouring countries, or countries from that area. Although there is general awareness of this problem (e.g. low hygiene and frequent scabies in camps), the same factor can be connected with the situation in specific countries and the increase in the frequency of scabies."

RESPONSE: We have followed this helpful suggestion and included a new section in discussion as follows, with eight new references [33-40] (340-358 in tracked version, inserted text in [square brackets]):

'[The wider context of scabies across Europe

Some studies suggest scabies infestations may have increased across Europe recently, including research conducted in Turkey, France and the Netherlands, countries in which a number of our study participants were based.[33–38] In the general population of the Netherlands recorded scabies diagnosis per 1,000 persons per year increased fourfold 2011–2020 (0.6 to 2.6), in parallel with a sixfold increase in scabicide dispensing over the same period. Most dispensations were permethrin; however large nosocomial outbreaks were linked to a peak of ivermectin prescriptions in 2014–2015.[34] Similarly, German studies found increasing numbers of cases since 2009, most notably an elevenfold increase in persons 15–24 years of age from 2009–2018.[35, 36, 38] In Croatia, where reporting of scabies is mandatory, general incidence increased sixfold 2007–2017, and multiple outbreaks were reported from adult care and nursing homes.[37] Some data suggests an association between population movement and higher incidence in the general population. However, though refugee/migrant groups have been disproportionately affected by scabies when living in the crowded camps which are the subject of our study, they are unlikely to be the cause of changes in reported overall prevalence within the wider population.[33, 35–38] Suggested explanations given for this claimed trend include increased number of sexual contacts, treatment failure in younger patients with presumed poorer compliance, and aging populations.[33, 34, 36–40] The observed trend may also be in part a result of reporting bias, given the higher attention scabies is now being given in some countries following its adoption as a WHO NTD.]'

The new references are:

[33] Bener F. Increase in scabies incidence: a retrospective cohort study. *Eu Res J* 2021;7:5. doi: 10.18621/eurj.770849

[34] van Deursen B, Hooiveld M, Marks S, et al. Increasing incidence of reported scabies infestations in the Netherlands, 2011–2021. *PLoS One* 2022;17(6):e0268865. doi 10.1371/journal.pone.0268865

[35] Delaš Aždajić M, Bešlić I, Gašić A, et al. Increased Scabies Incidence at the Beginning of the 21st Century: What Do Reports from Europe and the World Show?. *Life* 2022;12(10):1598. doi: 10.3390/life12101598

[36] Reichert F, Schulz M, Mertens E, et al. Reemergence of scabies driven by adolescents and young adults, Germany, 2009–2018. *Emerg Infect Dis* 2021;27(6):1693. doi: 10.3201/eid2706.203681

[37] Lugović-Mihić L, Aždajić MD, Filipović SK, Bukvić I, Prkačin I, Grbić DŠ, Ličina ML. An increasing scabies incidence in Croatia: A call for coordinated action among dermatologists, physicians and epidemiologists. *Zdr Varst* 2020; 59(4): 264–272. doi: 10.2478/sjph-2020-0033

[38] Sunderkötter C, Aebischer A, Neufeld M, et al. Increase of scabies in Germany and development of resistant mites? Evidence and consequences. *J Dtsch Dermatol Ges* 2019;17(1):15–23. doi: 10.1111/ddg.13706

[39] Zhang W, Zhang Y, Luo L, et al. Trends in prevalence and incidence of scabies from 1990 to 2017: findings from the global Burden of disease study 2017. *Emerg Microbes Infect* 2020;9(1):813-816. doi: 10.1080/22221751.2020.1754136

[40] Mbuagbaw L, Sadeghirad B, Morgan RL, et al. Failure of scabies treatment: a systematic review and meta-analysis. *Br J Dermatol* 2023;ljad308. doi: 10.1093/bjd/ljad308'

Thank you again for your supportive review, and helpful suggestion which we are sure will have improved our manuscript.

REVIEWER 2

RESPONSE: We are grateful for your positive evaluation of our study and your description of it as “an important article which raises the issue of an ad hoc approach to a common problem of scabies in refugee camps in Europe.” Thank you.

We appreciate the time you have put into considering and commenting on our manuscript, and we list below, point-by-point, the changes we have made in response.

R2: “I think it raises the issue of guidelines for standardised diagnosis and treatment. The only issue I think should be improved is perhaps drawing more attention to the IACS guidelines.”

RESPONSE: Thank you for this helpful steer. We have drawn more attention to them as follows: (1) we have added new text referring to them by name in the first paragraph of the introduction in lines 76-82 in tracked version (see our related response to reviewer 1 above for the inserted text). We have also added text into the diagnosis section of the discussion as follows (363-366 in tracked version, inserted text in [square brackets], deleted text strikethrough):

‘However, subsequently published [(2018)] consensus criteria for scabies diagnosis [developed by IACS (<https://www.controlscurabies.org>)] [34] provides a standard for clinical diagnosis[. This is summarised in Box 1, and fully detailed in Engelman et al.,[42]] though it requires validating in these [refugee/migrant camp] settings.’

Finally, we felt rather than summarising them ourselves (which may obscure carefully written consensus agreed wording), we instead would include the following Box in our manuscript, which is taken from Engelman et al [42] (which was published under CC BY 4.0) with full attribution (line 470 in tracked version):

‘[Box 1. Summary of 2018 IACS criteria for the diagnosis of scabies

A: Confirmed scabies

At least one of:

A1: Mites, eggs or feces on light microscopy of skin samples

A2: Mites, eggs or feces visualized on individual using high-powered imaging device

A3: Mite visualized on individual using dermoscopy

B: Clinical scabies

At least one of:

B1: Scabies burrows

B2: Typical lesions affecting male genitalia

B3: Typical lesions in a typical distribution and two history features

C: Suspected scabies

One of:

C1: Typical lesions in a typical distribution and one history feature

C2: Atypical lesions or atypical distribution and two history features

History features

H1: Itch

H2: Close contact with an individual who has itch or typical lesions in a typical distribution

Notes

1. These criteria should be used in conjunction with the full explanatory notes and definitions (in preparation).
2. Diagnosis can be made at one of the three levels (A, B or C).
3. A diagnosis of Clinical and Suspected scabies should only be made if other differential diagnoses are considered less likely than scabies.

Attribution: Reprinted from Engelman et al (2018),[42], which is licenced under CC BY 4.0 (<https://creativecommons.org/licenses/by/4.0>). No changes have been made.]’

R2: "Also, in the Pacific, very good results of ivermectin MDA have been achieved without any environmental measures on hygiene or disinfection of bedding etc. This should also be pointed out."

RESPONSE: We have followed your instruction and added the following text in discussion, with two new references 53 and 54 (lines 404-407 in tracked version, inserted text in [square brackets]):

'[It should however be noted that (primarily) ivermectin-based MDAs have achieved very good results in community intervention trials across Oceania (for a summary, see [53]) without any environmental measures on hygiene or disinfection of bedding etc. (for example, see work in the Solomon Islands [54]).']

New references:

'[53] Middleton J. Can ivermectin mass drug administrations to control scabies also reduce skin and soft tissue infections? Hospitalizations and primary care presentations lower after a large-scale trial in Fiji. *Lancet Reg Health - West Pac* 2022;22: 100454. doi: 10.1016/j.lanwpc.2022.100454

[54] Romani L, Marks M, Sokana O, et al. Efficacy of mass drug coadministration of ivermectin and azithromycin for control of scabies and impetigo: a single-arm community intervention trial. *Lancet Infect Dis* 2019;19:510–518. doi:10.1016/S1473-3099(18)30790-4'

Thank you again for your helpful review which we are sure has improved our manuscript.

(Added 28/9/23 RESPONSE EDITORIAL ASSISTANT COMMENTS on 25/9/23)

EA: '1. Funder Grant Number

- You have indicated a funder/s for your paper. Please ensure to provide an award/grant number for your funder/s in the main document file and in ScholarOne.
- If the funder cannot provide an award/grant number, you can indicate N/A for the award/grant number.'

RESPONSE: In the main document file we have added 'N/A' after 'Brighton and Sussex Medical School from internal funds' (lines 496-497 on tracked version). I have also added 'N/A' in ScholarOne.

EA: '2. Figures. Kindly rename your uploaded figures to figure 1 and figure 2.'

RESPONSE: We have done as instructed and renamed the uploaded files as figure 1 and figure 2.

EA: '3. Supplementary Tables. The in-text citation for 'Supplementary Table 2' is missing. Please provide the missing citation and ensure that all citations of supplementary tables are in ascending order.'

RESPONSE: As instructed, we have provided the missing citation (line 118 in tracked version), and also reordered sections in the supplementary file to make sure all citations of supplementary tables are in ascending order in the main text and in the same order in the supplementary file. This also necessitated updating some line and page numbers given in reporting checklists, which we have also done.